# Why Is Surgery Still Done after Concurrent Chemoradiotherapy in Locally Advanced Cervical Cancer in Romania?

**DOI:** 10.3390/cancers16020425

**Published:** 2024-01-19

**Authors:** Silviu Cristian Voinea, Cristian Ioan Bordea, Elena Chitoran, Vlad Rotaru, Razvan Ioan Andrei, Sinziana-Octavia Ionescu, Dan Luca, Nicolae Mircea Savu, Cristina Mirela Capsa, Mihnea Alecu, Laurentiu Simion

**Affiliations:** 1“Carol Davila” University of Medicine and Pharmacy, 050474 Bucharest, Romania; silviu.voinea@umfcd.ro (S.C.V.); rotaru.vlad@gmail.com (V.R.); andreiioanrazvan@gmail.com (R.I.A.); sinziana.ionescu@umfcd.ro (S.-O.I.); luca_dan94@yahoo.com (D.L.); mihnea.alecu@umfcd.ro (M.A.); laurentiu.simion@umfcd.ro (L.S.); 2General Surgery and Surgical Oncology Department II, Bucharest Institute of Oncology “Prof. Dr. Al. Trestioreanu”, 022328 Bucharest, Romania; 3General Surgery and Surgical Oncology Department I, Bucharest Institute of Oncology “Prof. Dr. Al. Trestioreanu”, 022328 Bucharest, Romania; 4General Surgery Department, “Sf. Ioan” Clinical Emergency Hospital, 042122 Bucharest, Romania; 5Radiotherapy Department, Bucharest Institute of Oncology “Prof. Dr. Al. Trestioreanu”, 022328 Bucharest, Romania; mrcsavu@yahoo.co.uk; 6Radiology Department, Bucharest Institute of Oncology “Prof. Dr. Al. Trestioreanu”, 022328 Bucharest, Romania; cristinacapsa@yahoo.com

**Keywords:** locally advanced cervical cancer, concurrent chemoradiotherapy, adjuvant surgery, local control, high cervical cancer morbidity, residual disease, cervical cancer in Romania

## Abstract

**Simple Summary:**

Romania ranks second in Europe in both mortality and incidence rates for cervical cancer, and the majority of cases are diagnosed in locally advanced stages. Therapeutic guidelines indicate concomitant chemoradiotherapy as the proper treatment (total dose 85–90 Gy at point A). Yet, in Romania, as a result of poor radiotherapy infrastructure, deficient patient access to radiotherapy, and specific socio-psychologic factors, concurrent chemoradiotherapy is usually suboptimal (total dose 60–65 Gy at point A) and often followed by radical surgery aimed at better local control. Our retrospective cohort study of 351 patients with histopathological confirmed locally advanced cervical cancer proved that complete pathologic response is present in less than half of patients previously treated by concurrent chemoradiotherapy, thus demonstrating the rationale behind radical surgery—better local control, especially in unfavorable aggressive histological types like adenocarcinoma and adenosquamous carcinoma or tumors that have progressed under concurrent chemoradiotherapy. We discuss several possible causes for the over-usage of surgery in Romania for treating locally advanced cervical cancer.

**Abstract:**

The incidence and mortality of cervical cancer are high in Romania compared to other European countries, particularly for locally advanced cervical cancer cases, which are predominant at the time of diagnosis. Widely accepted therapeutic guidelines indicate that the treatment for locally advanced cervical cancer consists of concurrent chemoradiotherapy (total dose 85–90 Gy at point A), with surgery not being necessary as it does not lead to improved survival and results in significant additional morbidity. In Romania, the treatment for locally advanced cervical cancer differs, involving lower-dose chemoradiotherapy (total dose 60–65 Gy at point A), followed by surgery, which, under these circumstances, ensures better local control. In this regard, we attempted to evaluate the role and necessity of surgery in Romania, considering that in our study, residual lesions were found in 55.84% of cases on resected specimens, especially in cases with unfavorable histology (adenocarcinoma and adenosquamous carcinoma). This type of surgery was associated with significant morbidity (28.22%) in our study. The recurrence rate was 24.21% for operated-on patients compared to 62% for non-operated-on patients receiving suboptimal concurrent chemotherapy alone. In conclusion, in Romania, surgery will continue to play a predominant role until radiotherapy achieves the desired effectiveness for local control.

## 1. Introduction

Globally, cervical cancer ranks fourth in terms of incidence and mortality. In Romania, it represents a significant public health issue due to the elevated rates of incidence and mortality, among the highest (ranking second) in Europe [1,2]. Like other gynecologic cancers, the rates are higher in Central and Eastern Europe than in the rest of the continent [3].

There are several causes responsible for this unfavorable situation. Firstly, there is no national screening program for cervical cancer, which often leads to late-stage diagnosis. Secondly, there is a lack of an active and sustained vaccination campaign against HPV for the female population. Lastly, it is relatively difficult to access radiation centers [4]. As a consequence, cervical cancer ranks third among all neoplastic localizations in Romania in terms of incidence [1].

While there is no formal definition for the term “locally advanced cervical cancer,” there is a broad consensus that includes stages IB2 (AJCC 2017)/IB3 (FIGO 2018)-4A [5]. These stages are considered to carry an increased risk of local recurrence and/or metastasis [6] and thus require more aggressive treatment [7].

All current therapeutic guidelines consider that the treatment for locally advanced cervical cancer should be concurrent chemoradiotherapy [8,9]. An exception is made for stages IB3 and IIA2, for which radical surgery (hysterectomy) is proposed either alone or after concurrent chemoradiotherapy [8]. However, in Romania, it is common practice to perform surgery after concurrent chemoradiotherapy in locally advanced cervical cancer, which significantly deviates from the unanimously accepted therapeutic guidelines (ESGO/ESTRO/ESP or NCCN) [4,10]. These guidelines recommend definitive radiotherapy (external radiotherapy + radio-sensitizing chemotherapy + brachytherapy) with a total dose of 80–90 Gy at point A. It is worth noting that the current medical practice in most Romanian centers for treating certain conditions involves administering external irradiation with a total dose of 45–50 Gy, alongside weekly doses of cisplatin at 40 mg/m^2^ (5 administrations), followed by brachytherapy (or not) at a dose of only 15 Gy. Thus, the total dose at point A amounts to merely 60–65 Gy, well below the standard set by therapeutic guidelines. However, in recent years, a small number of centers (mostly private) have started to offer image-guided brachytherapy and even MRI-guided brachytherapy at optimal doses. After 6–8 weeks of chemoradiotherapy, cases that respond well and become operable (partial or complete responders) undergo surgery. For these cases, radical hysterectomy or extra fascial hysterectomy with pelvic lymphadenectomy is performed, with or without lymph-node sampling or latero-aortic lymphadenectomy [4,10].

## 2. Materials and Methods

We conducted a retrospective cohort study at the Institute of Oncology “Prof. Dr. Al. Trestioreanu”, the oncology center with the highest volume of diagnosed and treated cases of cervical cancer in Romania. The study was conducted in accordance with the Declaration of Helsinki and approved by the Institutional Review Board (or Ethics Committee) of Bucharest Institute of Oncology “Prof. Dr. Al. Trestioreanu” (protocol code: 11193, date of approval: 29 August 2023). The study includes 351 patients with histopathological confirmed cervical cancer stages IB3–IB, according to FIGO staging criteria, treated between January 2015 and December 2021 by the same multidisciplinary team and according to the same investigations and treatment protocol, evaluated from a total of 383 cases. All patients underwent the following preoperatory work-up: chest X-ray/thoracic CT scan, pelvis and abdomen MRI/PET-CT scan/CT scan, and cystoscopy/colonoscopy for suspicion of bladder or recto-sigmoid invasion. The treatment protocol consisted of whole-pelvis external-beam radiation (50.4 Gy), cisplatin (40 mg/m^2^/weekly for 5 days), and brachytherapy to point A (15 Gy). After 4–6 weeks, another MRI or PET-CT of the pelvis and abdomen was performed for the re-evaluation in preparation for surgical intervention. After 6–8 weeks, a laparotomy was followed by type C2 (type III) radical hysterectomy or extra fascial total hysterectomy with pelvic lymphadenectomy with or without lateral-aortic lymph node sampling. In cases where there was an intraoperative pelvic or latero-aortic lymphatic invasion, only an extra-fascial total hysterectomy was performed, and these cases were not abandoned. During the same period, a total of 58 patients with locally advanced cervical cancer underwent the same concomitant chemoradiotherapy but experienced unfavorable progression of the disease locally or the development of distant metastases. These patients either underwent laparotomy or received no surgical intervention, leading to therapeutic abandonment. Clinical and imaging follow-up were the only approaches. Laparoscopic approaches, the ICG-guided sampling of pelvic lymph nodes, and other types of modified surgical procedures, although described as possibilities in the literature, were not indicated for our patients due to the technical particularities of the cases [11,12,13].

The surgical resection specimens were evaluated for pathologic response, which was classified as either partial (pPR) or complete (cPR). The patient data were analyzed by grouping them into subsets based on their histologic tumor type. We also analyzed the outcomes of each case in terms of recurrence, lymphatic and distant metastasis, and surgery-related morbidity.

### Statistical Analysis

The statistical analysis was performed using the NCSS 2019 software. A two-proportions comparison test was employed to assess the impact of histologic tumor type on complete pathologic response (cPR) and pathological complete regression (PCR). To address potential imbalances in clinical variables among the three tumor categories (squamous carcinoma, adenocarcinoma, and adenosquamous carcinoma), we applied a 1:3 propensity score (PS) matching technique. The primary objective of this technique was to diminish inadequate matches between the aforementioned categories. Statistical significance was determined if the *p*-value was less than 0.05. The data are presented as numerical counts and percentages.

## 3. Results

Most of our cases were classified upon the histopathologic examination of resected specimens as squamous cell carcinomas—301 (85.75%). The rest were adenocarcinomas—28 (7.98%), adenosquamous carcinomas—19 (5.41%), and neuroendocrine tumors—3 (0.85%).

The complete pathological report of surgical specimens found that in 196 cases (55.84%) the result of the concurrent chemoradiotherapy was a partial pathologic response (pPR) and a residual tumor in the cervix was present (Figure 1). Less than half of the patients (44.16%) showed no residual malignant cells on the resected specimen. In 70 cases (19.94%), the pathologic examination revealed a concomitant residual tumor of the cervix, as well as pelvic lymph-node invasion (53 patients—15.1%) and latero-aortic lymph node invasion (17 patients—4.84%). In the 196 cases with pPR after concurrent chemoradiotherapy, 139 cases (70.92%) showed a less than 50% reduction of pre-therapeutic tumor size, and 50 cases (25.51%) had a pathologic response greater than 50%, with 7 (3.57%) of those having only a microscopic residual tumor in the cervix (Figure 2).

It is interesting to note that cases of adenocarcinoma and adenosquamous carcinoma, which are more aggressive tumors, exhibit a higher percentage of residual tumors compared to cases of squamous carcinoma (Figure 3). This finding is consistent with the existing literature [14]. 

Additionally, cases with adenocarcinoma and adenosquamous carcinoma have a much lower rate of complete pathological response (pCR) compared to cases with squamous cell carcinoma.

In surgically treated patients, procedure-related complications were seen in 101 cases (28.77%), including intraoperative incidents and accidents, early complications (first 7 days postoperatively), and late complications. There were major complications in 26 cases (7.40%), which included urinary fistulas, complex vesico-vaginal or recto-vaginal fistulas, bowel obstructions, and peritonitis due to bowel perforation (Figure 4). 

The recurrence rate in surgically treated patients was 24.21% (85 cases), including locoregional recurrence (pelvic or latero-aortic) and distant metastases (lymphatic, osseous, hepatic, pulmonary) (*p* < 0.05). The 58 patients with locally advanced cervical cancer who underwent the concomitant chemoradiotherapy but experienced an unfavorable progression of the disease locally or the development of distant metastases were either subjected to laparotomy or no surgical intervention was performed, essentially leading to therapeutic abandonment, with clinical and imaging follow-up being the only approach. Typically, these patients are not included in any studies in Romania and fall into a gray area. In this particular group, the recurrence rate was significantly higher (36 cases, 62.1%; *p* < 0.05), including local, regional, and distant metastatic recurrences.

## 4. Discussion

It has not been proven that surgery is beneficial after concurrent chemoradiotherapy. It is now widely recognized that this type of surgery does not improve overall or disease-free survival compared to definitive irradiation. However, the postoperative morbidity, particularly urinary tract morbidity, is significantly higher [15,16,17]. Retrospective studies and meta-analyses have shown similar survival after concurrent chemoradiotherapy, with or without subsequent surgery [16,17,18,19,20,21]. Other smaller retrospective studies even suggest a benefit of surgery after concurrent chemoradiotherapy in terms of overall survival (OS) and local disease control [22], or that it may improve disease-free survival (DFS) without affecting overall survival, specifically for patients with good responses to concurrent chemoradiotherapy [23].

However, after concurrent chemoradiotherapy, residual lesions at the cervical level persist in a variable proportion, ranging from 32 to 60% across different stages [15,18,24,25], as also illustrated by our study. The exact prognostic significance of these residual lesions after concurrent chemoradiotherapy remains uncertain, as they are not synonymous with recurrence but do result in significantly worse outcomes; various studies have shown that residual cervical lesions lead to a higher rate of local recurrence [26,27]. Concurrent chemoradiotherapy should ensure local control within therapeutic guidelines.

A serious issue arises in patients who do not respond to concurrent chemoradiotherapy or experience disease progression. These patients are left with suboptimal doses and are only monitored clinically and via imaging techniques. Thus, they remain undertreated according to current standards. This category of patients, who are therapeutically abandoned and not included in any evaluation or study in Romania, presents an oncological challenge due to the high rate of therapeutic failure, consequently leading to unfavorable outcomes in the majority of cases.

Regarding the assessment of treatment response to chemoradiotherapy in locally advanced cervical cancer patients, 18F-FDG PET/CT seemed to exhibit superior diagnostic accuracy compared to other imaging modalities. MRI has shown a notably low sensitivity in identifying metastases, but PET/CT demonstrated considerably superior performance. Nevertheless, there was no substantial disparity seen in the diagnosis of residual illness between these two approaches, as was explained by Sistani [28].

According to research by Fleischmann [29], there is a lack of a wide range of molecular markers that can be used to accurately predict how patients will respond to medication and how long they will survive. Additionally, there is a need to identify people who have either a high or low risk of developing certain conditions. Nevertheless, the use of these indicators might enhance treatment outcomes and facilitate the development of novel targeted medicines. As an example of these previous statements, a study by Du [30] underlined that ERCC1 has been identified as a very promising biomarker for cervical cancer. Analysis of ERCC1 polymorphism suggests that it might serve as a valuable tool for predicting the likelihood of developing cervical cancer and the potentially hazardous side effects of therapy. Another marker that has proven useful in predicting tumor response to neoadjuvant treatment is a high pretreatment level of squamous cell carcinoma antigen (SCC Ag), which is linked to large tumors and a low chance of survival in patients with cervical cancer who undergo definitive concurrent chemoradiotherapy (CCRT). Physicians may use SCC Ag levels to inform their decision-making process for surgery, hence minimizing the risks associated with dual treatment techniques. An increased level of SCC Ag is linked to resistance to radiation, and the pace at which SCC Ag decreases during concurrent chemoradiotherapy (CCRT) may serve as an indicator of tumor response after treatment, as demonstrated also by Fu [31].

As stated also by Rosolen [32], microRNAs (miRNAs) have a role in controlling mitochondrial activity and maintaining balance, as well as influencing cell metabolism. They do this by targeting certain oncogenes and tumor suppressor genes that are part of metabolic-related signaling pathways associated with the fundamental characteristics of cancer. The functions of miRNAs and their respective target genes have mostly been documented in cellular metabolic processes, mitochondrial dynamics, mitophagy, apoptosis, redox signaling, and resistance to chemotherapeutic drugs. Collectively, these results confirm the involvement of miRNAs in the metabolic reprogramming characteristic of cancer cells and emphasize their potential as predictive molecular indicators for therapy response and/or targets for therapeutic intervention.

The contraction of HPV may lead to the development of cancer, and conventional therapies often lead to the reappearance of the disease. The validation of liquid biopsy using HPV circulating tumor DNA (HPV ctDNA) as a potential diagnostic for predicting recurrence in HPV-related malignancies has yet to be determined. HPV infection has the potential to induce cancer, and conventional therapies often lead to relapse. The validation of liquid biopsy using HPV circulating tumor DNA (HPV ctDNA) as a potential indicator for predicting recurrence in HPV-related malignancies is still pending, according to the conclusion of a study by Karimi [33].

Cuproptosis has been implicated in the process of carcinogenesis and the advancement of cancer. Nevertheless, the clinical effects of long non-coding RNAs (lncRNAs) associated with cuproptosis (CRLs) in cervical cancer (CC) are not well understood. Research was conducted in order to discover novel biomarkers that can accurately forecast prognosis and assess the effectiveness of immunotherapy, with the ultimate goal of enhancing this scenario. An analysis was conducted to assess the potential efficacy of the prognostic signature in predicting the response to immunotherapy and the susceptibility to chemotherapeutic medicines. Kong [34] generated a risk signature consisting of eight long non-coding RNAs (lncRNAs) associated with cuproptosis (AL441992.1, SOX21-AS1, AC011468.3, AC012306.2, FZD4-DT, AP001922.5, RUSC1-AS1, AP001453.2) to predict the survival outcome of patients with cervical cancer. We then assessed the reliability of this risk signature. Utilizing our 8-CRLs risk signature, we conducted an independent evaluation of the effectiveness and responsiveness to immunotherapy in patients with cervical cancer. This signature has the potential to enhance clinical decision-making for personalized treatment.

As explained by Lakomy [35], the immunological interactions involved in the genesis, progression, and therapy of cervical cancer are intricate. Compelling evidence supports the notion that lymphopenia and an increased neutrophil-to-lymphocyte ratio are predictive of unfavorable outcomes, while other markers also show possible connections. In the context of aggressive medical treatments, there is also an important matter that concerns people with various degrees of immunodepression—for instance, infection with HIV. Connected to this theme, the majority of studies examined by Shah [36] indicated that there were no disparities in treatment outcomes, such as overall toxicity, treatment response, or death, based on HIV infection status.

Concerning the “direct monitoring” of the response to treatment, a paper by Schernberg [37] showed that adaptive radiation is based on the ability to monitor changes in the structures of target volumes in order to make adjustments to the treatment plan during radiotherapy. This approach considers both internal movements and the reaction of the tumor. The use of MRI technology in radiotherapy linear accelerators has made it possible to monitor motion during the administration of treatment. MRI may also be used to precisely assess the remaining volume of cervical tumors after chemoradiotherapy, enabling personalized treatment planning for brachytherapy boosts while taking into account the tumor’s radiosensitivity.

Outcome prediction models may assist in making relevant therapeutic choices due to the many variables that might predict the response of cervical cancer to therapy. The prediction models for cervical cancer toxicity, local or distant recurrence, and survival provide promising outcomes with an acceptable level of prediction accuracy, as presented by Jha [38].

Taking all this into account, the simplest answer to “Why is surgery still done after concurrent chemoradiotherapy in locally advanced cervical cancer, in Romania?” is that it is used to assess the response to suboptimal concurrent chemotherapy practiced in the country and to achieve better local control. However, it is known that this type of surgery does not improve survival when performed after guideline-stipulated concurrent chemoradiotherapy and is associated with significantly increased additional morbidity, as shown by our study. However, in Romania, the radiation doses used in current practice by most centers are less than those stipulated. There are several reasons why irradiation is not performed according to international therapeutic guidelines in Romania.

First is the reluctance of radiation therapists to encounter complications from definitive irradiation (rectal/vesical/vaginal fistulas, extensive post-radiation fibrosis, and post-radiotherapy enteritis with subsequent obstructions), although these complications also occur after the irradiation practiced in our country, leading patients to seek surgical solutions. As a result, definitive irradiation is rarely carried out in Romania.

Another challenge is the limited access that patients have to radiation centers. Although Romania has several such centers, they are mostly concentrated in a few university centers, causing delays in getting enrolled for irradiation. As a result, only a small percentage of patients have easy access to treatment. Additionally, there is evidence suggesting that the quality of irradiation is uneven across different centers, leading to unpleasant post-irradiation complications that are difficult to manage. The poor radiotherapy infrastructure in Romania, coupled with the usage of old equipment, leads to inequitable access to therapy for patients, with physicians altering prescribed dosage and intervals in order to work around the often-malfunctioning equipment and perform miracles in order to facilitate access for most patients. According to the DIRAC (Directory of Radiotherapy Centers), in 2023, Romania had 38 radiation centers (state and private hospitals/centers), with a total of 82 mega-voltage radiotherapy units and only 10 brachytherapy units. All the equipment is concentrated in the largest cities in Romania (over 50% being in the capital city, Bucharest, Cluj-Napoca and Iasi). About a third of all radiation units are 10 years old or older, and two units are over 40 years old. Brachytherapy is available in only four cities (Bucharest, Cluj-Napoca, Iasi and Oradea) [39]. Sometimes, due to a lack of access to radiation therapy or suboptimal neoadjuvant therapy, surgery is used as a replacement, leading to the overuse of surgery to achieve better local control, especially in cases of locally advanced cervical cancer. Residual lesions can be surgically removed for better local control after concurrent chemoradiotherapy, according to studies [15,24,40].

Thirdly, the socioeconomic conditions, particularly in Romania, which has an average level of development, do not allow for the performance of all tests and investigations required to ensure proper follow-ups for all patients, as described by other studies [41]. Due to economic conditions and low health literacy, Romanians often delay preventive health measures, leading to many patients being lost from observation—for example, the vast majority of cervical cancer patients, after going through concurrent chemoradiotherapy followed by surgery, do not return for regular follow-ups.

Finally, there is a specific psychological-cultural factor for Romanians that also plays a role in the overuse of surgery after concurrent chemoradiotherapy. Many patients believe that being operated on gives them a better chance; they perceive surgery as the only treatment that can remove the cancer from the body. This is a common belief among our patients. Sometimes, physicians also share this belief (mostly due to frustration with the current diagnostic and therapeutic possibilities available to them and to the knowledge that many patients have low health literacy and do not realize the dangers of not having regular follow-ups after cancer therapy).

Also, we asked ourselves if we could replace the surgical assessment of definitive chemoradiotherapy with a less invasive method. The alternative, imaging studies, does not allow for a very accurate assessment. After concurrent chemoradiotherapy, MRI is the most useful tool for assessing residual lesions due to its high predictive value and low false-negative rate [42,43]. However, there is a discrepancy between the control pelvic MRI (performed after concurrent chemoradiotherapy) and the pathologic response evaluated on the resection specimen, which in our case, occurred in 38 patients.

It is a fact that in Romania, we perform this type of surgery after concurrent chemoradiotherapy. The question to answer is: What are its indications and in which cases is such surgery useful? Currently, in Romania, clinical and imaging assessments are conducted at 4–6 weeks, and the gynecologic oncologist decides if the intervention can be technically performed or not, as described in our study. However, coherent and reasoned indications for this type of surgery should exist, even though presently they do not. In our opinion, surgery should only be performed for cases with a complete clinical and imaging response post concurrent chemoradiotherapy. For these cases, performing surgery has been suggested, but only a total extra-fascial hysterectomy accompanied by pelvic lymphadenectomy is proposed to reduce postoperative morbidity, by avoiding the parametrial direction of the ureter [43,44]. Another possibility would be surgery after concurrent chemoradiotherapy for cases without clinical and imaging response or those with disease progression, which could improve prognosis by excising larger lesions [25]. Another approach would be surgery for cases with unfavorable histopathology, i.e., cases with adenocarcinoma or adenosquamous carcinoma, known for their more aggressive progression [14] and in which our study also showed a very high percentage of residual cervical lesions post concurrent chemoradiotherapy. In cases like these, a weak response to treatment is highly likely, so adjuvant surgery may offer better local control. Recently, for advanced cases (stage IVa or lower but associated with complex fistulas between the pelvic organs), extensive surgery has sometimes been indicated when the resection is technically possible—pelvic exenterations have been performed for such cases with encouraging results [45,46], in some cases even having radical curative intent. 

The final question to answer is: How can we determine the utility of this type of surgery for locally advanced cervical cancer? Given the current therapeutic approach in Romania and the possibilities and availability of diagnostic and therapeutic modalities, there is a need for large prospective studies to compare outcomes in terms of survival and complication rates with and without adjuvant surgery. Another approach, as described by Querleu [47], would involve more effective technical irradiation or new cytostatic drugs for better radio sensitization or the implementation at a national level of standardized procedures for concurrent chemoradiotherapy—methods that would increase the effectiveness of neoadjuvant therapy and make adjuvant surgery unnecessary or significantly reduce its use.

In order to mitigate the over-usage of surgery after concurrent chemoradiotherapy in Romania, there is a profound need for restructuring and improving the radiotherapy infrastructure available. Efforts from all decision-makers and interested social parties (including non-profit organizations and citizens) should be aimed at establishing a truly regional network of radiotherapy centers with new centers being developed in smaller cities. Also, there is a profound need for awareness campaigns addressed to patients aimed at improving health literacy and changing high-risk behaviors.

## 5. Conclusions

While not yet confirmed conceptually, the practice of performing surgery after concurrent chemoradiotherapy is currently being utilized in Romania. In the context of suboptimal radiotherapy, surgery aims to achieve better local control at the cost of increased morbidity. The rationale behind the usage of surgery is that it can provide a level of certainty about the pathologic response after concurrent chemoradiotherapy. Our research found that over 50% of excised specimens contained residual lesions, even in the absence of an evident tumor, on the imaging evaluation performed after neoadjuvant therapy. Surgery can provide better control of tumors that have progressed under concurrent chemoradiotherapy or have unfavorable histology, which are known for their aggressivity. Also, surgery is used as a substitute to substandard follow-up and for symptom control in locally invasive cervical cancers.

The poor status of radiotherapy infrastructure in Romania—uneven distribution of radiation centers, concentration of equipment in large cities, old age of radiation units, and non-standardized treatment courses—together with specific socio-psychologic of the Romanian population—low health literacy that influences patients’ health decisions, resulting in difficult follow-up—are the principal causes that lead to the over-usage of surgery after concurrent chemoradiotherapy in the treatment of locally advanced cervical cancer. As long as the conditions of radiotherapy in Romania do not improve and measures are not taken to better educate the population about health issues and the importance of adequate health behavior, surgery will continue to maintain a dominant position in the treatment of locally advanced cervical cancer.

## Figures and Tables

**Figure 1 cancers-16-00425-f001:**
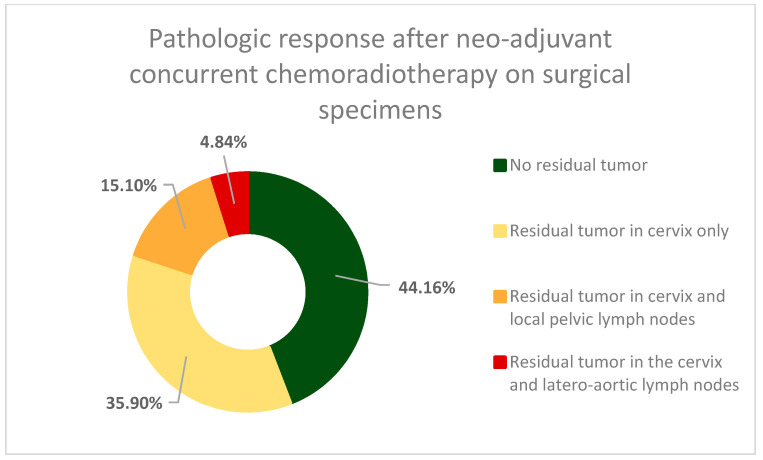
The distribution of cases according to pathologic response after concurrent chemoradiotherapy—All surgical specimens underwent complete histopathological analysis, and the cases were stratified. In 155 cases (represented in green) complete pathologic response was observed and there was no evidence of residual tumor on the resected specimens and the cases were classified as ypT0N0. In 196 cases malignant elements were found on the resected surgical specimens (partial pathologic response) and were classified as ypT1-2N1-2—in 2/3 of these cases the residual tumor was located in the cervix only, and in the rest lymph node involvement was also present.

**Figure 2 cancers-16-00425-f002:**
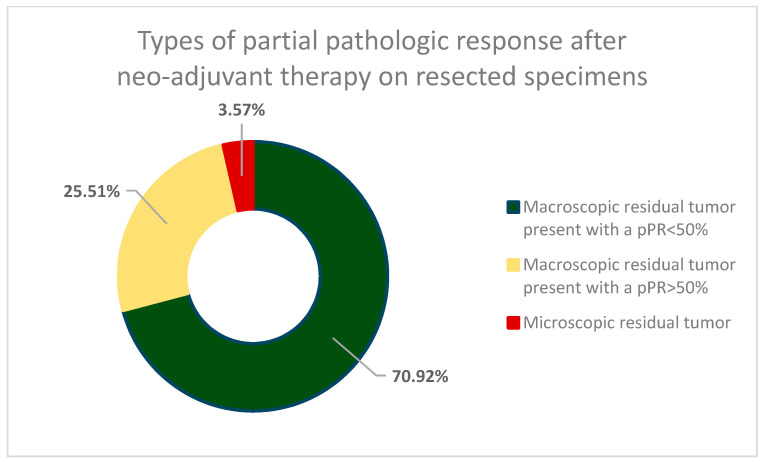
Types of partial pathologic response (pPR) encountered on resected surgical specimens. Majority of pPR cases were found to have macroscopic residual tumor (96.43%) and over 70% presented a pathologic response of less than 50% when compared to tumor size before neoadjuvant concurrent chemoradiotherapy.

**Figure 3 cancers-16-00425-f003:**
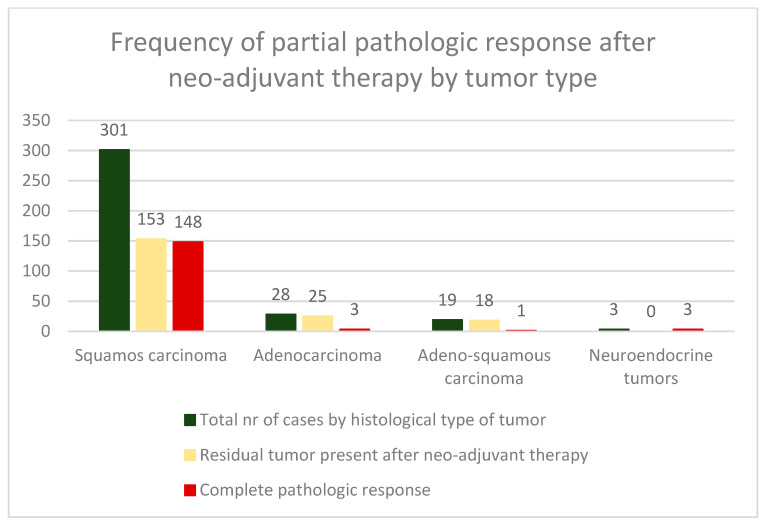
Frequency of complete/partial pathologic response after neoadjuvant concurrent chemoradiotherapy by histopathological type of tumor. The presence of residual malignant elements on surgical resection specimens is more frequent in tumors with aggressive histology like adenocarcinoma and adenosquamous carcinoma (*p* < 0.05).

**Figure 4 cancers-16-00425-f004:**
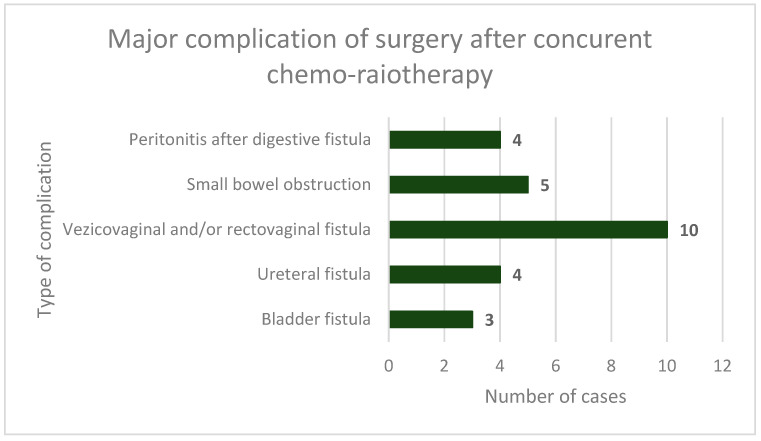
Major complications of surgery performed after neoadjuvant chemoradiotherapy (including intraoperative incidents, early and late complications)–frequency and type.

## Data Availability

Since it involves personal data, due to privacy issues, data will be available upon request by e-mail to C.I.B.

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
