# Peer review of "Why Is Surgery Still Done after Concurrent Chemoradiotherapy in Locally Advanced Cervical Cancer in Romania?"

_cancers, 2024, doi:10.3390/cancers16020425_

Round 1

Reviewer 1 Report

Comments and Suggestions for Authors

In the article "Why is surgery still done after concurrent chemo-radiotherapy in locally advanced cervical cancer in Romania?" Voinea et al. analyzes splints from a retrospective group with locally advanced cervical cancers trying to explain why surgery is still proposed after chemo-radiotherapy. Knowing very well the situation in Romania, I would recommend the authors to include some additional details.

1. 15Gy single fraction brachytherapy to point A is not currently the standard in Romania. There are centers that offer image-guided brachytherapy and even MRI-guided brachytherapy at optimal doses.

2. The settlement of brachyteharpy services is actually a cause of the underdevelopment of this dethiba at present and the lack of accessibility.

3. The specific psychological-cultural factor is also involved - the patient believes that if he is operated he has an extra chance

The introduction is too long and the discussion about biomarkers should be moved to another separate subchapter.

The presented data could be imported and solutions must be proposed to provide the optimal solution in a country where cervical cancer occupies a worrying place for a country that wants to align itself with developed countries. I also recommend the revision of the expression and the evaluation by a native English speaker

Author Response

Dear Reviewer,

First, we the authors, would like to express how deeply honored we are that our work was submitted to such a rigorous and extremely patient reviewer, which also has a deep knowledge of the Romanian oncologic system. We are delighted that you have dedicated your time to an in-depth analysis of our manuscript and we thank you for your suggestions, which allowed us to improve our work and produce a relevant material. After reviewing our material in accordance with your suggestions, we also found a few additional small mistakes that we were able to rectify in time, and we are grateful for your contribution.

Second, you made some very reasonable points, to which we adhere. They were omissions in our part and we revised the text per your suggestions, rewritten and extended some paragraphs. Point by point we addressed the suggestions as follows:

  1. “15Gy single fraction brachytherapy to point A is not currently the standard in Romania. There are centers that offer image-guided brachytherapy and even MRI-guided brachytherapy at optimal doses”

We rectified our expression which can indeed be construed as us stating that this is the standard for Romania. We added rows 75-77 of the revised manuscript explaining that even if most centers provide substandard or no brachytherapy, we’ve seen in the last years the emergence of various centers that offer advanced image-guided optimal dosed procedures.

  1. “The settlement of brachytherapy services is actually a cause of the underdevelopment of this dethiba at present and the lack of accessibility”.

We discussed the poor radiotherapy infrastructure in Romania in rows 397-417 of the revised manuscript and its role in the low accessibility to radiotherapy services of most patients – availability, distribution and age of equipment.

  1. “The specific psychological-cultural factor is also involved - the patient believes that if he is operated he has an extra chance”

We added rows 425-438  explaining the role this factor plays.

  1. “The introduction is too long and the discussion about biomarkers should be moved to another separate subchapter”.

We have shortened the introduction and moved the discussion about biomarkers in the Discussion section

  1. “The presented data could be imported and solutions must be proposed to provide the optimal solution in a country where cervical cancer occupies a worrying place for a country that wants to align itself with developed countries”.

Rows 478-484 of the revised manuscript have been added proposing a few solutions for mitigating the over-usage of surgery after concurrent chemo-radiotherapy in Romania.

  1. “I also recommend the revision of the expression and the evaluation by a native English speaker”

The manuscript was subjected to multiple automated and manual grammar and spelling checks and it also has been revised by one of our authors which holds a Native Speaking Level English Language qualification. If deemed necessary, we will use professional checking by mdpi for our manuscript before printing.

We hope that our explanations are satisfactory. For further clarifications we remain at your disposal.

Reviewer 2 Report

Comments and Suggestions for Authors

The manuscript by Voiena et al. discusses about the importance of surgery in Romania even after chemoradiotherapy for locally advanced cervical cancer.

However, some suggestions to improve the manuscript are:

1. Some of the references are not indexed properly, needs correction.

2. The conclusion section should be re-written with more details.

3. The quality of figures should be improved and the figure legends should be elaborated.

Author Response

We would like to thank you for your valuable comments. All of your suggestions have been analyzed and we corrected the errors of reference indexing, we have re-written the conclusion section and added more details as requested and we also tried to adjust the figures of our manuscript.

We hope the modifications are to your satisfaction and we remain at your disposal for any further questions.

Round 2

Reviewer 1 Report

Comments and Suggestions for Authors

In this version the article could be published

Author Response

We thank you for your kind remarks and patient review of our manuscript.

Reviewer 2 Report

Comments and Suggestions for Authors

1. The figure legends are still not up to the mark, needs to be further elaborated.

2. In the figure legend of Figure 3, "p<0.05" should be mentioned instead of "p<005".

Author Response

We thank you for your very pertinent remaks. We have redone all of the figures in the articles improving resolution and readability. The legends of figures were re-written and further elaborated. The mistake you pointed out in the legend of Figure 3 was corrected and we also corrected a similar mistake in the text of article.

Hoping we have addressed your concerns we remain at your disposal.

Round 3

Reviewer 2 Report

Comments and Suggestions for Authors

The revised version of the manuscript looks better.